# Implicit neural obfuscation for privacy preserving medical image sharing

**Mattias P. Heinrich**[1]                    MATTIAS.HEINRICH@UNI-LUEBECK.DE
[1] *Institute of Medical Informatics, Universität zu Lübeck, Germany*

**Lasse Hansen**[2]                                LASSE@ECHOSCOUT.AI
[2] *EchoScout GmbH, Lübeck, Germany*

**Editors:** Accepted for publication at MIDL 2024

## Abstract

Despite its undeniable success, deep learning for medical imaging with large public datasets leads to an often overlooked risk of leaking sensitive patient information. A person's X-ray, even with proper anonymisation applied, can readily serve as fingerprint and would enable a highly accurate re-identification of the same individual in a large pool of scans. Common practices for reducing privacy risks involve a synthetic deterioration of image quality, e.g. by adding noise or downsampling images, before sharing them publicly. Yet, this also adversely affects the quality of downstream image recognition models trained on such datasets. We propose a novel strategy for finding a better compromise of model quality and privacy preservation by means of implicit neural obfuscation. Our method jointly overfits a neural network to a small batch of patients' X-ray scans and applies a substantial compression - the number of network parameters representing the images is more than 6x smaller than the original images. In addition, we introduce a k-anonymity mixing that injects partial information from other patients for each reconstruction. That way identifiable information is efficiently obfuscated, while we manage to maintain the quality of relevant image parts for the intended downstream task. Experimental validation on the public RANZCR CLiP dataset demonstrates improved segmentation quality and up to 3 times reduced privacy risks compared to a more basic image obfuscation baselines. In contrast to other recent work that learn specific anonymous representations, which no longer resemble visually meaningful scans, our approach remains interpretable and is not tied to a certain downstream network. Source code and a demo dataset are available at https://github.com/mattiaspaul/neuralObfuscation.

**Keywords:** neural implicit representation, anonymisation, obfuscation, image sharing

## 1. Introduction / Motivation

The trend towards larger models, in particular vision transformers, for image recognition have exemplified the need for training with millions of images at the same time. While the advent of grand challenges in medical imaging has led to an ever increasing amount of public CTs, MRIs and X-rays - their amount is still orders of magnitudes smaller than natural image databases (e.g. LVD-142M or SA-1B). Yet, hundreds of millions of digitised scans (Schöckel et al., 2020) are acquired and stored in local clinical picture archives each year. The vast majority of them is never shared (anonymously) with the research community, one likely strong reason being privacy concerns and tighter regulations (Mostert et al., 2016). Despite its benefits of restricting direct access to personal information the current process of image anonymisation or pseudonymisation is far from perfect (Kaissis et al.,

2020). (Packhäuser et al., 2022) revealed a severe risk of re-identification *even if rigorous anonymisation of images is performed*, which may enable an attacker to find a person with probabilities as high as 90% within a large public dataset given another X-ray of them. In fact millions of scans together with medical reports have already been leaked due to poor IT security at some hospitals[1] that could be linked to anonymised data and increase the risk of re-identification attacks even further. Our objective is hence to devise a safer mechanism that enables anonymous image data release with substantially reduced re-identification risk, but at the same time this data should retain its diagnostic value for a given intended downstream task, e.g. semantic segmentation.

## 2. Related work

Much research has been devoted to de-identifying individuals in natural images or video sequences. Since visual re-identification risks pose a severe challenge to comply with current data privacy regulation obfuscation strategies have been devised to modify images to make persons harder to identify. The DP-Net (Fan, 2018) explores blurring, black/white boxes as well as adversarially learned degradations (cf. also (Wu et al., 2018)) to maintain the targeted downstream task performance while reducing privacy leakage. (Zhu et al., 2020) and (Dall'Asen et al., 2022) propose to create synthetic image replacements (DeepFakes for de-identification) to preserve privacy in medical videos while preserving diagnostic features for downstream tasks, i.e. preserving keypoints. Advanced methods for video-based person re-identification have been developed in (McLaughlin et al., 2016). (Kim et al., 2021a) and (Packhäuser et al., 2023b) proposed to learn certain geometric deformations that make the re-identification of brain MRI or chest X-rays with retrieval learning much harder. Latent diffusion models are explored in (Packhäuser et al., 2023a) to create replica datasets that demonstrate only moderate performance drops for training models for downstream abnormality classification, while enhancing privacy preservation. (Kim et al., 2021b) propose a Privacy-Net that jointly learns to map input MRI brain scans into an intermediate privacy-preserving representation, train a semantic parcellation U-Net and also minimises the re-identification risk. While showing excellent results for the given tasks, this procedure requires access to paired patients for each annotation (which is often not fulfilled) and leaves the intermediate representations not interpretable for humans. Mixup-privacy (Kim et al., 2023a) is another strategy aimed at avoiding full knowledge transfer between client and server. Both can therefore be more closely associated with recent differential privacy approaches in federated learning (Rieke et al., 2020) that could also be supplemented by encryption with mathematical security guarantees (Kaissis et al., 2021). k-anonymity, which mixes information from several identities in a single output datapoint, can be seen as a particularly promising strategy to strike a good balance between privacy preservation, downstream task performance and interpretability of the obfuscation. (Meden et al., 2018) compare several approaches for k-anonymity including k-Same-Pixel (Newton et al., 2005) and a new proposed k-Same-Net along with basic pixelation strategies for face photos. They demonstrate good performance for learning to generate synthetic images that share attributes from multiple persons but are specific labels (age, gender, facial expression).

---

1. https://www.blackhat.com/eu-23/briefings/schedule/index.html#millions-of-patient-records-at-risk-the-perils-of-legacy-protocols-34188

**Contribution:** Our method advances the state-of-the-art in effective medical image obfuscation strategies with regards to the following three main points:

- robust generative model, by adapting recent work on neural implicit representation and compression for video sequences to the obfuscation of a subset of an X-ray collection,

- novel strategy for k-anonymity that only moderately affects visual image quality while substantially reducing re-identification risks, and

- alleviation of the strong requirements of prior work that are based on simultaneous availability of multiple scans per patients at each data provider

Along with these technical contributions, we advance the field of privacy concerning medical deep learning with comprehensive experiments that include the evaluation of privacy risks along downstream task performance (semantic segmentation of catheters in X-rays) for baselines compared to our proposed model. Furthermore, we provide reproducible code for public Kaggle challenge data for others to replicate and built upon our work.

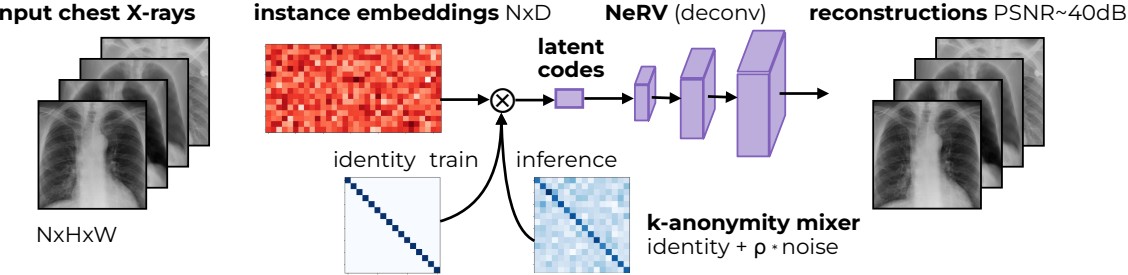

Figure 1: Concept figure of proposed implicit neural obfuscation strategy. A number of input chest X-rays serve as target for a neural reconstruction decoder that comprises learnable instance embeddings (D-dimensional vector for each data point) and convolutional weights. The reconstructions are supervised with a loss based on structural image similarity (SSIM). During inference a k-anonymity mixing is introduced that aims to obfuscate patient information by adding latent code information from other patients.

## 3. Methods

Our study comprises three aspects: image obfuscation, semantic X-ray segmentation and siamese network re-identification. The concept is implemented within the following scenario. Several data providers want to contribute anonymised X-ray scans along with detailed expert annotations of clinically relevant objects. Here, we use pixel level segmentations of foreign material, in particular central venous catheters (CVC), which are commonly used to detect critical malpositioning (Roldan and Paniagua, 2015). We assume that part of the

combined dataset comprises images with the same patient pseudonym that can be used to train a siamese retrieval network, which will be used to assess the re-identification risk. But crucially neither every image has to be annotated with CVC labels nor does every patient have to be present multiple times. Hence, we do not assume the possibility of jointly training an image obfuscation strategy to de-identify patients along with the segmentation task but rather require the obfuscation to work as a stand-alone step. In addition and in contrast to (Kim et al., 2021b) and (Packhäuser et al., 2023b), we define the obfuscation strategy to be a white-box model that is accessible to the potential attacker, since having to keep such methods hidden to the public while sharing them across multiple clinics would pose another severe risk/challenge. Our main contribution lies in the development of a novel strategy for creating partially k-anonymous scans using neural implicit compression for open data sharing that preserve relevant feature to train semantic segmentation networks. Yet, the employed semantic segmentation and siamese re-identification methods are described as well for completeness.

**Implicit neural obfuscation:** We base our work on the recent NeRV approach for neural representations for video compression (Chen et al., 2021). Implicit Neural Representations (INRs) are rapidly gaining attention for effective image representations that amongst others enabled performance leaps for 3D reconstruction (Mildenhall et al., 2021), image compression (Strümpler et al., 2022) or alignment (Lin et al., 2021; Wolterink et al., 2022).

The key observation is that a low parameterisation of a fully-connected or convolutional network is sufficient to represent images based on an input of a positional encoding. Extending INRs to larger datasets (e.g. through amortised learning (Sitzmann et al., 2020)) is not trivial, yet several newer approaches either employ learnable encoders (Kim et al., 2023b) to predict a latent code embedding for each image or simply keep a dictionary of embedding vectors. (Chen et al., 2021) implements the latter and learns a compact decoder model to restore a video sequence. They clearly demonstrate that in contrast to traditional auto encoders, which have a shared encoder for the whole dataset, NeRV improves reconstruction quality by training a new model for each subset (in their work short video clip). For our approach, we adopt this concept and fit a NeRV to each chunk of 64 images in our data set. We specify the decoder to start from a 64-dimensional latent vector that is mapped with a fully-connected layer into a 16-channel $3 \times 3$ latent code and then upsampled with convolutions and pixel-shuffle operations to a target image size of $360 \times 360$ pixels. We firstly experimented with a mean-squared error reconstruction loss (used traditionally in auto-encoders to mimic a maximum likelihood model) yet this led to unsatisfactory results. Minimising the structural dissimilarity index (maximising SSIM) (Woods et al., 1998), however, achieves high quality reconstructions with good convergence. The concept is presented in detail in Fig. 1.

Next, we introduce a k-anonymity mixer into the inference path of our NeRV-image reconstruction. A $N \times N$ matrix, which is the sum of an identity and Gaussian noise with a hyperparameter $\rho$ controlling the standard deviation, is multiplied with the instance embeddings. That way the latent codes share information from other patients in the same mini-batch. Because the noise is injected at the lowest level of the convolutional decoder it also affects global contextual image content and will ideally mask a substantial amount of

identifiable information. This step is only performed at inference, once a subset of images has been fully fitted to avoid the risk of learning to reintroduce personal fingerprints.

**Catheter segmentation:** We opt to use semantic segmentation of catheters as downstream task, due to its clinical relevance paired with challenges for obfuscated images. Central venous catheters are extremely thin foreign objects that typically form an elliptic curve that end in the vena cava. We employ a straightforward 2D SegResNet model (using the MONAI implementation) (Myronenko, 2019). A unit-weighted combination of soft Dice loss and binary cross entropy (after sigmoid activation) is used to train the network with pixellevel supervision. Note, that we always assume high-quality annotations are available and do not deteriorate labels as they pose a very limited risk for re-identification.

**Re-identification** We implement a classic siamese re-identification network (Taigman et al., 2014) that comprises two identical ResNet34 streams, which produce $D$-dimensional feature encodings for each image within a mini-batch of size $N$. A cosine similarity is applied to produce a $2N \times 2N$ score matrix which is fed into the objective function, noise-contrastive estimation loss (InfoNCE) (Oord et al., 2018), which aims to maximise the similarity of the only positive example out of each $2N - 1$ candidates.

## 4. Experiments and Results

The data was obtained from the Kaggle RANZCR CLiP challenge[2]. We follow a similar pre-processing as (Hansen et al., 2021) in that we first predict lung masks to each X-ray and automatically define a suitable bounding box for each scan. The images and labels are resampled to $384 \times 384$ pixels and the CVCs are dilated to approx. 5 pixels. The whole dataset comprises >10'000 images, but we make a subselection to datapoints that either contain a normal CVC annotation or a part of a patient that occurs at least twice to be able to evaluate the re-identification risk. This yields 1536 training and 512 test scans for CVC segmentation and 576 patients with 1152 scans - and 384 patients with 768 images respectively for training and testing for the re-identification risk evaluation (note: the sets do not have to be disjoint).

For the image reconstruction/obfuscation, we leave the architectural design setup as is based on the public NeRV repository [3], yet we substantially decreased the capacity of the model to avoid overfitting. In initial experiments, we aimed for approx. 500k trainable parameters per batch of 128 images, which yields a compression of >33-fold when assuming the same quantisation of model weights and image pixels and image dimensions of $360 \times 360$ pixels. However, this resulted in an under-fitting of the reconstructed images with missing details. Hence, we opted for tripling the parameter count and storing 64 images per NeRV, which is still a considerable 6-fold compression and yields PSNRs of, on average, approx. 40dB. Such a high agreement with the original data will obviously make the re-identification of the same person easier and hence decrease the desired privacy preservation. It is therefore crucial to adjust a suitable noise parameter $\rho \in \{0, 0.04, 0.06, 0.08\}$ for the proposed k-anonymity mixing.

---

2. https://www.kaggle.com/c/ranzcr-clip-catheter-line-classification/data
3. github.com/haochen-rye/NeRV

As baseline obfuscation strategy, we employ pixellation. That means a range of compressed versions of the input images are obtained by downsampling the input images by factors of $\{1, 2, 4, 6\}$ and resampling them afterwards to the original resolution. We expect both the segmentation quality and re-identification risk of models trained with these degraded images will be lowered.

For the 2D SegResNet we chose 24 initial feature channels and 3.5 million parameters in total. The model is trained with a batch-size of 32 for 375 epochs (number of training images is 1536). We use Adam with an initial learning of $2 \cdot 10^{-3}$ that is reduced by half every 1500 iterations and restarted every 4500 iterations. The first 1000 iterations are stabilised with an additional heatmap loss. We employ the `RandomPhotometricDistort` and `RandomErasing` augmentation from Torchvision (v2) and add affine geometric transformations (with a standard deviation of $7 \cdot 10^{-2}$ and random horizontal flipping to both images and labels. At test time, we only include the horizontal flip, hence two predictions are averaged per input.

For training a siamese re-identification network, we follow common practices of contrastive self-supervised learning and use an Imagenet-pretrained ResNet34 for each batch of 32 image pairs. The output feature size is fixed to 256 channels and the InfoNCE loss with cosine similarity and a temperature of $7 \cdot 10^{-2}$ is used as loss. Adam was used with initial learning rate of $10^{-3}$ for 444 epochs with a single step of 0.2 at half-time. The same augmentations are used as before for the SegResNet. This time, we also include them for testing as we found otherwise all models could not cope with the large diversity of scanning parameters and/or geometric misalignment. We average 25 predictions for each pair of potential matches. Having 384 patients in the test set, we compute the risk of re-identification for a single other image of that same person in the training using any number of guesses from 1 to 15, meaning e.g. the random chance at top-5 would be about 1%.

For both experiments (segmentation and re-identification) we evaluated whether the models trained with obfuscation perform better (here higher re-identification means better even though this is a worse outcome for an algorithm) with original test scans or the same modulation. We found that using obfuscated images at test time works in all instances best, likely due to the fact that the models have learned to adapt to these images. Crucially, all re-identification attacks reported were also retrained with the same obfuscation strategies to adapt to the knowledge of the defence mechanism. All models are trained on RTX A4000 cards with bfloat16 precision and `torch.compile` - within a typical run time of 1 hour. Further implementation details can be found in our public source code at: https://github.com/mattiaspaul/neuralObfuscation.

**Segmentation Results:** Apart from the strongest pixellation variant, all approaches perform reasonably well with average Dice scores of above 75% on the challenging downstream task. Remarkably the NeRV obfuscation with $\rho = 0.08$, which introduces strong visible artefacts still produces predominantly high quality segmentations as visualised in Fig. 2 and 4. The highest overall performance of 86% is reached by using the original images followed by NeRV without k-anonymity and half-resolution scans with each 83%.

**Re-identification Evaluation:** Putting the segmentation scores into context with privacy preservation, it becomes evident that only the strongest pixellation with poor downstream performance comes close with a top-5 risk of 47% to the lowering of re-ID risks of

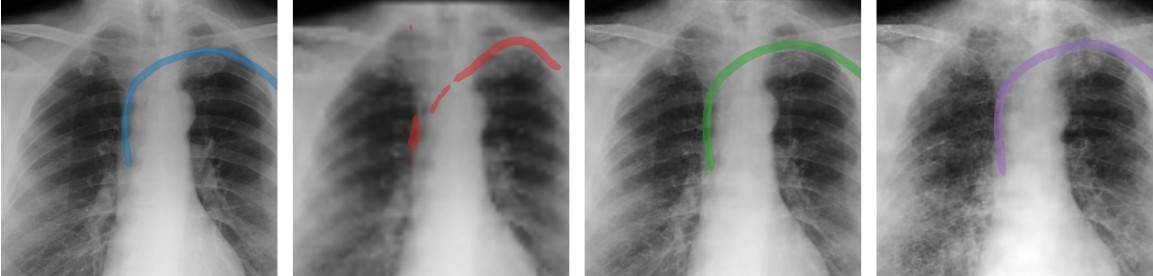

Figure 2: Visual result comparing both obfuscation and downstream performance. From left to right: ■ original image with ground truth segmentation; ■ pixellation to $\frac{1}{6}$ resolution; ■ NeRV based obfuscation with $\rho = 0.04$ and; ■ $\rho = 0.08$ respectively. Clearly, the neural obfuscation better balances personal de-identification and diagnostic quality. More results are found in the appendix.

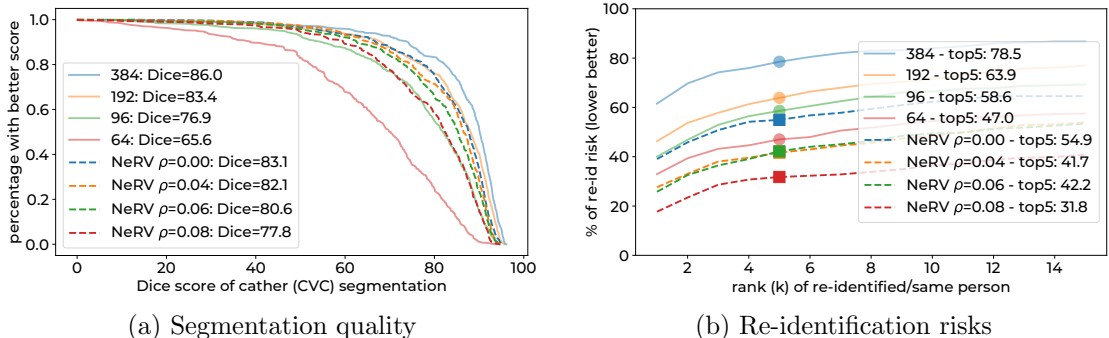

(a) Segmentation quality          (b) Re-identification risks

Figure 3: Comparison of cumulative statistics of segmentation quality vs. re-identification risk. NeRV obfuscation with $\rho = 0.04$ is on par for CVC segmentations while posing a 50% lower privacy risk (at top-5) as pixellation with half-resolution (indicated as 192).

the implicit neural obfuscation. When increasing $\rho$ from 0.04 to 0.08 the top-5 risk decreases from 42% to 32%. The contrast is particular stark for top-1 re-identification with original images, >60% compared to our NeRV with $\rho = 0.08$ with <20% - a three-fold improvement. Choosing $\rho$ depends on the intended use case and privacy-risk assessment. Since pre-trained task- and re-identification models can be quickly evaluated with different $\rho$s for a given NeRV model this provides a first indication of suitable choices. A re-training of both networks is, however, required to validate this assessment.

## 5. Discussion and Conclusion

Our study demonstrates that privacy risks are imminent for anonymous medical image data sharing, but they could be addressed by a suitable neural obfuscation strategy with negligible performance drops of the models trained and evaluated with such data. The white-box model leads to interpretable outputs and does not impact the process of training downstream models, since normal images can be shared. It is computationally lightweight requiring on average one second to fit a NeRV per image (one minute for a batch of 64). The compression of $> 6$-fold also brings benefits for a more efficient data transfer. This is the first time neural implicit obfuscation is used for interpretable X-ray segmentation and the proposed introduction of k-anonymity yielded a large improvement in risk reduction.

**Limitations:** There are limitations with regards to the employed comparative methods, since we restricted them to be viable in a scenario where not all labelled data has to be available with multiple scans per patient during training. In case this is possible, even stronger performance could be achieved by specifically optimising de-identification together with segmentation. We also wanted to avoid black-box obfuscation models that have to be kept secure for further anonymisation steps, e.g. at another clinical centre. This is not strictly necessary if all data comes from one hospital. Further initial experiments to extend the number of baseline comparisons to deformation based obfuscation and mixup-privacy as well as the extended evaluation on our NeRV based approach to digitally reconstructed radiographs (DRRs) and another downstream segmentation task can be found in our Github repository and Supplementary material.

**Future work:** It is not yet clear, how such an approach could be extended to sharing volumetric scans. 3D CTs and MRI comprise substantially more anatomical detail and could thus lead to even greater privacy risks. There are also other promising strategies to learn implicit image embeddings, e.g. using generalisable INRs (Kim et al., 2023b). While being more complex to train, using meta-learning, they could decouple larger parts of the neural networks between shared and instance based parameters and hence provide more control over the level of compression and obfuscation. It also remains to be seen, whether a dataset with NeRV-based k-anonymity still excels at other tasks of chest pathology detection.

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

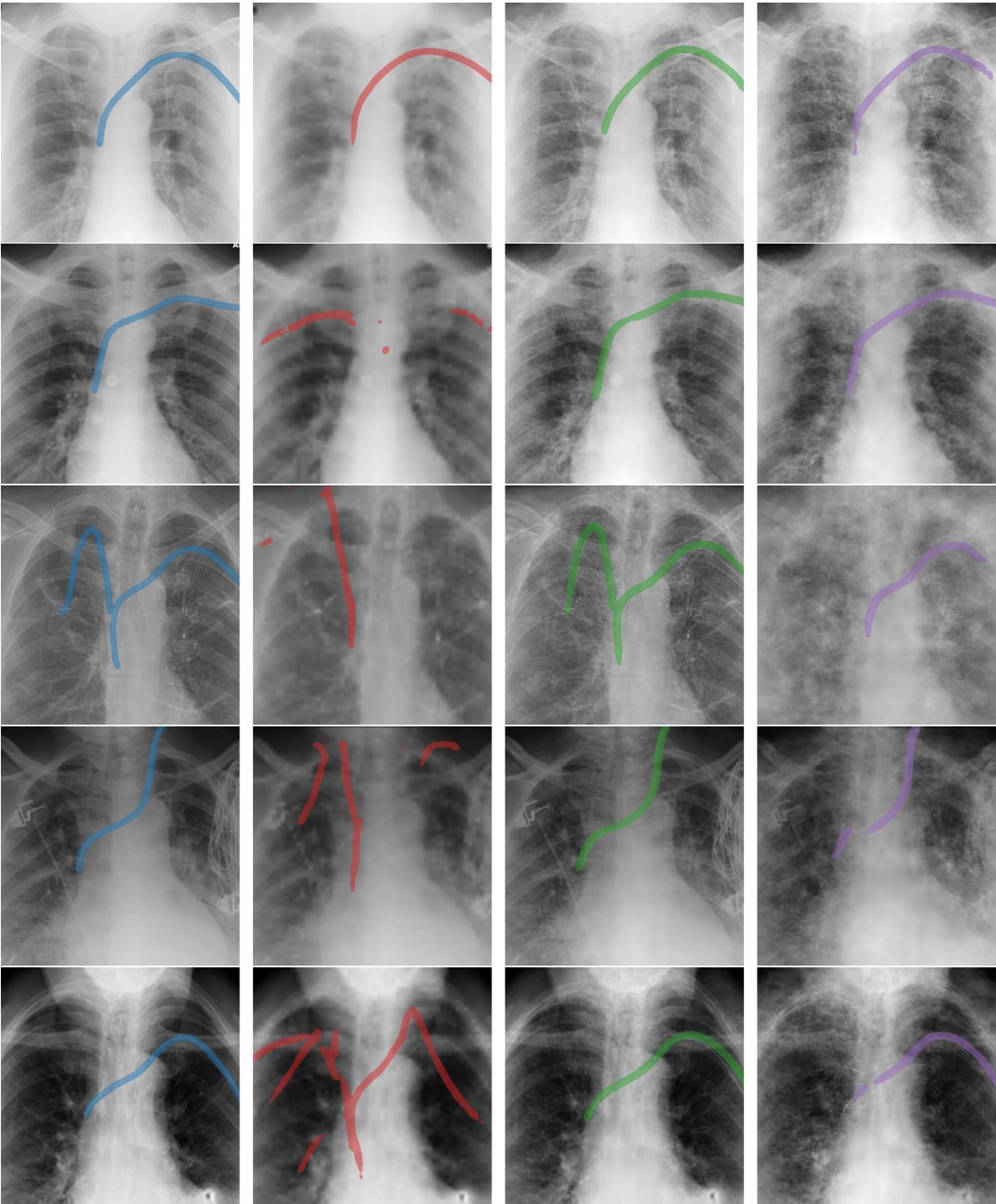

Figure 4: Additional results comparing both obfuscation and downstream performance. From left to right: ■ original image with ground truth segmentation; ■ pixellation to $\frac{1}{6}$ resolution; ■ NeRV based obfuscation with $\rho = 0.04$ and; ■ $\rho = 0.08$ respectively.

