# OpenReview forum: "Implicit neural obfuscation for privacy preserving medical image sharing"
_MIDL.io/2024/Conference — MIDL 2024 Poster_

### Official Review · Reviewer_iTVz · 2024-02-26

**Confidence:** 4
**Preliminary Rating:** 1
**Final Rating:** 2

**Summary:**

The paper presents a novel approach for privacy-preserving medical imaging based on learning implicit neural representations. In this approach, a neural network with few parameters overfits a small batch of images on the task of reconstruction. The learning neural representations, which correspond to a compressed version of images, are mixed to generate noisy reconstructions. In experiments, the paper that these noisy reconstructions offer a better protection of identity (i.e., it is harder for a Siamese network to determine to co-ownership of two reconstructed images) while also enabling a good segmentation.

**Strengths:**

* The paper introduces an interesting and novel use of neural representation learning, which has fewer limitations than adversarial obfuscation and reduces the costs of data transfer. It also offers a simple way to control the trade-off between obfuscation and task usefulness (hyperparameter rho)

* The paper is generally well written

* The authors provide source code for their method

**Weaknesses:**

* The experimental validation of the method lacks depth: the method is only compared against an very weak baseline (pixellation) and on a single dataset/task/modality.

* The end application of this method is not clearly presented in the paper. If the goal is simply to transfer images, then encryption offers stronger guaranties in terms of privacy. On the other hand, if the objective is training a centralized model, then the model is still susceptible to leaking patient identity information since the segmentation output (which is sent back to the client is not encoded). Finally, if the model is to be used on a federated learning setup, then the paper should compare against approaches of this type.

* I have some serious doubts about the re-identification results of the method. If I understand correctly, the Siamese re-identification network is trained on original images, hence it may suffer from domain shift when used on reconstructed images. As such, the re-identification score might largely underestimate the possibility of recovering patient identity from reconstructed images. Looking at the images in Fig 3, it seems quite easy to recognize the subject in reconstructed images (although longitudinal data should be shown to confirm this).

* The section on Related work is missing important papers on the topic (see comments below)

**Detailed Comments:**

Other comments:

* p2: (Kim et al., 2021) propose a Privacy-Net that jointly learns to map input MRI brain scans into an intermediate privacy-preserving representation, train a semantic parcellation U-Net and also minimises the re-identication risk...  this procedure requires access to paired patients for each annotation (which is often not fulfilled)..." [In  (Kim et al., 2023), the same authors proposed a method that does not have this requirement and is also based on mixing]  ... makes the obfuscation model a new point of vulnerability. [In (Kim et al., 2021), the image encoder is conditioned on a private key, thus removing this vulnerability].

Kim BN, Dolz J, Jodoin PM, Desrosiers C. Mixup-Privacy: A simple yet effective approach for privacy-preserving segmentation. In International Conference on Information Processing in Medical Imaging 2023 Jun 8 (pp. 717-729). Cham: Springer Nature Switzerland.

Kim BN, Dolz J, Desrosiers C, Jodoin PM. Privacy Preserving for Medical Image Analysis via Non-Linear Deformation Proxy. BMVC 2021

* p5: "due not have" --> do not have

* Figure 3. The caption should better explain that 384, 192, 96 and 64 correspond to different downsampling sizes of the pixellation method (if I understood correctly)

**Justification Of Final Rating:**

I acknowledge the potential of the proposed method but feel that this current version of the paper falls short of expectations. Authors have performed necessary experiments but present them in a shallow way (in an Appendix). The visual results in Figure 2 cast doubt on the level of privacy offered by the proposed method.

Therefore, I give a final score of weak reject.

**Justification Of The Preliminary Rating:**

* The proposed method is novel and interesting. However, the experimental validation does not properly demonstrate the advantages of this method with respect to recent approaches in the literature, as well as on different tasks/datasets/modalities.

* Important related works are missing

* There is a potential flaw with the re-identification results in the paper, or these results are not properly described in the paper.

**Questions To Address In The Rebuttal:**

Clarify the end application setup of your method and explain how your experimental evaluation is fair (with respect to this setup). Moreover, assuming that the decoder is shared, I am not sure that the encoded images offer the level of protection claimed by the paper. Authors should further demonstrate this claim.

---

> ### Author Response · Authors · 2024-03-18
>
> We thank the reviewer for their very comprehensive remarks and appreciation of the novelty of the method and are confident that our response can address many of the remaining concerns.
>
> Regarding the fairness of the experimental evaluation we apologise for our unclear wording of the following sentence: "For both experiments (segmentation and re-identification) we evaluated whether the models trained with obfuscation perform better (here higher re-identification means better even though this is a worse outcome for an algorithm) with original test scans or the same modulation". We meant to **highlight that we indeed re-trained the re-identification networks with knowledge of the obfuscation strategy**. This is now clarified as "Crucially, all re-identification attacks reported were also retrained with the same obfuscation strategies to adapt to the knowledge of the defence mechanism." in the same paragraph.
>
> We would like to re-iterate the differences of our intended use case of our method for privacy-enhanced public data sharing, which is not directly comparable to the very comprehensive methods presented in the literature e.g. by Kim et al. for obfuscation in a (at least partially) federated learning scenario. To answer one question from the weakness paragraph: Yes, the objective is training a centralised model without encryption. Given that the anatomies and foreign objects depicted in chest radiographs are less privacy-revealing than e.g. structures in brain MRI we hence consider the leakage of patient identity information through segmentations as rather low.
>
> We are very thankful for the two additional references (Kim et al. IPMI and BMVC) and **have incorporated both of them in our revised paper**. We **have also removed the statement about model vulnerability** with regards to the encrypted encoding in Kim et al. 2021. As suggested and detailed in the summary response, we included additional comparison experiments for deformation obfuscation and mixup-privacy. These experiments should not be interpreted as a re-implementation of the corresponding IPMI and BMVC papers, because the scope and intended use case is quite different. But they show that: 1) deformation obfuscation is another complementary solution that could further reduce privacy risk, but could be slightly diminished by re-training the attack with simulated warps; 2) mixup privacy works best in the scenario of a joint centralised model training and client obfuscation with appropriate hand-over of the information pertaining the mixup strategy. A straightforward sharing of mixed-up images and labels seems to be at least not easily possible for chest X-ray with ambiguous thin object segmentations.
>
> Further minor corrections are included in the revised PDF as suggested.

---

> > ### Comment · Reviewer_iTVz · 2024-03-25
> > **Response to the authors' rebuttal**
> >
> > I thank authors for considering all my comments and providing a detailed answer to my questions.
> >
> > Regarding my concern about the weak comparison baselines, authors have added two privacy-preserving approaches, based on on mixup and learned deformation. I apologize to the authors if I missed something but I do not see the mentioned Appendix in their revised paper or any Supplemental material document in the OpenReview page.
> >
> > For both added approaches, unlike the proposed method, the precise identity obfuscating transformation operation is not known to the attack because it depends on some information only known by the client. Is this how the approaches were used in the comparison or were the exact transformations shown to the attacker.
> >
> > To clarify my comment on the re-identification results, the reconstructed images in Figure 4 are visually similar to the original ones (especially for rho=0.04), and it seems easy to find identity using a simple correlation analysis (across a database of existing images). Could the author explain how identity is hidden here? Maybe I am wrong, but I assumed from the text that the image decoder is public hence encoded images can be reconstructed by a potential attacker.

---

> > > ### Author Response · Authors · 2024-03-26
> > >
> > > We thank the reviewer for considering our answers and would like to respond to the additional comments. The PDF was updated by us, which directly contains the appendix (we will ask the OpenReview support whether the version that authors see can be different). Nevertheless, the main findings are the ones explained in our direct answer: both added approaches, deformation and mixup, as well as our proposed method assume the same knowledge of a potential attacker about the obfuscation method. That means in particular for the deformation strategy that the attacker knows which range of displacements and B-spline model is used, **but does not know the exact transformations**.
> > >
> > > We hence stand by our statement that deformations can be an orthogonal strategy to further improve privacy preservation. However, they are on their own - for our experimental setup - less safe. With regards to the mixup strategy (proposed by Kim et al. at IPMI 2023), we would like to re-iterate that this in fact provides very strong privacy, but cannot achieve comparable segmentation accuracy in our chosen scenario due to our focus on data sharing without client-server interaction and a substantially different domain/downstream objective.
> > >
> > > We thank the reviewer for pointing to the visual similarity of different versions of the same patient/same scan in Fig. 4. Indeed a copy-detection would be able to retrieve the same scan for all considered obfuscation methods. The used setting for re-identification, however, requires the attacker to find another scan of that same patient within a database of openly shared images, which is much harder. **As suggested we evaluated a simple correlation analysis (measuring SSIM across compared scans), which revealed much lower re-identification scores** (approx. 10% top5 when each patient was present twice and 16% top15 accuracy when each patient was four times in the database). This demonstrates that the siamese retrieval network is more powerful and a reasonable choice to evaluate that indeed the obfuscation shown in Fig. 4 is strong enough to reduce re-identification risk for that patient.

---

> > > > ### Comment · Reviewer_iTVz · 2024-03-26
> > > > **Second response to authors**
> > > >
> > > > I thank authors once again for their responsiveness.
> > > >
> > > > Regarding the Appendix, I do see a long paragraph on page 12 however it seems to be a discussion of results which are not detailed in the paper. I expected authors to provide a table or a graph comparing their method to such baselines. I wish to point out that this section has no title (e.g., Appendix A) and mentions the following "..details of which can be found in the revised appendix.", making me believe that the Appendix is somewhere else.
> > > >
> > > > With respect to the re-identification results, I understand that the Siamese network has a low(er) accuracy however that does not necessarily mean that information related to identity is removed. As I mentioned in my previous comment, the reconstructed image for rho=0.04 is almost indistinguishable from the original one (visually speaking). I seriously doubt that clinical centres or regulatory agencies enforcing data privacy laws would be convinced by this low level of protection.

---

> > > > > ### Author Response · Authors · 2024-03-27
> > > > >
> > > > > We really appreciate the reviewers' remarks within this discussion because it highlights an important aspect that might have been overlooked in their assessment. There is an **important difference for deep learning based re-identification versus human identifiable similarity**. On the one hand, a strongly obfuscated image, e.g. by means of warping, can be impossibly difficult to re-identify by a person but easy for a machine that is trained to ignore (internally reverse) such "within-scan" obfuscation. On the other hand, the power of adversarial learning (and attacks) has demonstrated that subtle differences, which are inconceivable to the human eye may lead to drastic changes in decision boundaries. **Our contribution** - that was acknowledged as technically novel and interesting by all reviewers - **introduces subtle variations by mixing small details across patients, which are harder to reverse** for the siamese network as they stem from real other scans. This leads to an improvement in privacy-preservation, while maintaining a relatively high downstream task performance.
> > > > >
> > > > > We make no claim that this method is the only possible solution and superior to the work of Packhäuser or Kim et al., but refute the statement that regulatory bodies or privacy advocates in hospitals would consider this a "low level of protection", solely based on visual human perception of obfuscation strength - whereas our evaluation demonstrates the opposite. We believe conferences are a great possibility to discuss - even sometimes differing hypotheses - and would highly appreciate this opportunity.

---

### Official Review · Reviewer_Wxho · 2024-03-04

**Confidence:** 3
**Preliminary Rating:** 3
**Recommendation:** Poster
**Final Rating:** 4

**Summary:**

This paper proposes a privacy-preserving modification to implicit neural representation (INR). In particular, they consider INRs of a batch of images and multiply them with a Gaussian-noise perturbed identity matrix to anonymize the implicit representations. They show that representations learn this way preserve useful information, which is demonstrated by better performance on downstream segmentation task, and are less likely to leak private information (shown by poor reidentification of the subjects). They compare their approach with obfuscation technique such as pixelation or downsampling.

**Strengths:**

- Paper addresses an important problem of ensuring privacy preservation while releasing images/medical data.
- The method is interesting, easy to follow, and well-presented.
- They focus on data privacy, not neural network privacy, which may allow the safe release of datasets and not just the model.

**Weaknesses:**

### Guarantees
- The authors use an empirical measure of privacy by assessing reidentification risks. It would be nice to provide some mathematical quantification of privacy guarantees.
- What kind of "k-anonymity" is achieved could is unclear?

### Experiments
- The authors use only one empirical measure of privacy --- reidentification risks. Including other practical privacy measures to support the argument that leakage is reduced would be nice.
The proposed approach is compared to a very simple downsampling baseline. It would be nice to see comparisons with more advanced obfuscation techniques or even with differential private training techniques.

**Detailed Comments:**

See weaknesses.

Typos:
Abstract
- 6x smaller than the original pixels -> ... original images..?
Section 4:
- 10'000 -> 10,000

**Justification Of Final Rating:**

I am satisfied with the authors' answers to my questions and added new experiments and comparisons. The approach is interesting and effective; hence, I am increasing my score and would like this paper to be accepted.

**Justification Of The Preliminary Rating:**

The paper addresses an important problem, and the proposed solution is simple and interesting. However, the authors do not compare with more advanced baselines and lack mathematical or any other guarantee about the privacy of the data. This makes me less confident about the work. Hence, I would vote for borderline. I could lean towards accepting if these issues are fixed.

**Questions To Address In The Rebuttal:**

- How would method deal with streaming inputs at the time of inference? How should the other k-1 examples be picked and would that affect privacy guarantees?
- Fig 3a, what does % with better score (y-axis) mean?

**Special Issue:**

No

---

> ### Author Response · Authors · 2024-03-18
>
> We thank the reviewer for their appreciation of our solution for an important problem and great suggestions for further improving the validation. As mentioned in the general answer, we extended the set of comparison techniques by deformation-based obfuscation and mix-up privacy. While the former did not show as strong privacy preservation when retraining the re-identification network the latter led to a performance deterioration for the downstream task. We have corrected the typos and will add a short description of alternative solutions that could ideally be combined with our proposed method in future work.
>
> Following the recent PriMIA paper (Kaissis et al. 2023, https://doi.org/10.1038/s42256-021-00337-8) one could add homomorphic encryption and differential privacy when considering a federated learning setup - for which mathematical guarantees are available.
> To the best of our knowledge no universally agreed guarantees (aside from empirical tests) exist for obfuscated data sharing without encryption. But combining implicit neural obfuscation with DP (which has provable guarantees) could yield a beneficial defence against gradient-based model inversion attacks. Since our work does not focus on model sharing the latter do not directly apply for our experiments.
>
>
> Our answers to the two questions for the rebuttal are as follows.
> Streaming inputs at inference time could be handled in different ways: 1) by re-mixing a single new sample with images already obfuscated with our NeRV-solution (which would not affect privacy) or 2) if no further data sharing is intended (test-time) simply fitting the NeRV to a single image will also work based on our experiments.
> The y-axis in Fig. 3a) describes the cumulative nature of the Dice score plot. This means that e.g. at a threshold of 80% Dice (x-axis) the pixelation method with size 64 achieves a better score for just 25% of cases, whereas the original images (384) yield 85% of cases with Dice >80%.

---

### Official Review · Reviewer_QorR · 2024-03-04

**Confidence:** 4
**Preliminary Rating:** 4
**Recommendation:** Oral
**Final Rating:** 5

**Summary:**

In this work, the authors propose the use of implicit neural representation (INR) and k-anonymity mixer to enhance privacy preservation and decrease re-identification risks in a dataset of chest X-rays. To evaluate the approach, two experiments were performed: 1) catheter segmentation using a SegResNet, to assess the ability of the INR to preserve relevant features in the image despite compression and 2) re-identification using a siamese network to evaluate the re-identification risk. Compared to a conventional obfuscation approach such as pixellation, the approach showed better balance between segmentation quality and re-identification risk.

**Strengths:**

- The idea of using implicit neural representations to improve privacy preservation in medical data sharing is interesting and shows great potentials
- Methodology and experiments are well designed
- The paper is well written and the structure is appropriate
- Limitations of the study are noted and well discussed

**Weaknesses:**

- The experiments conducted are sufficient to demonstrate that the approach has potential, but it would be interesting to also test other downstream tasks, such as tumor detection or segmentation of other relevant structures in the image
- The choice of using only pixellation as baseline method can be limiting given the available promising deep-learning based approaches described in the Related work section
- Only a single dataset was used, it would be interesting to see the performance of the same approach with the same hyperparameters in representing images of other anatomical districts/other imaging modalities

**Detailed Comments:**

Only one minor comment: I recommend checking consistency of verbal tense usage, particularly in section Experiments and Results.

**Justification Of Final Rating:**

The authors responded clearly and concisely to all comments I raised in the review. Also, the authors followed my suggestion to implement and evaluate a second downstream task. I have no concerns with the last version of the manuscript, therefore, I strongly accept.

**Justification Of The Preliminary Rating:**

In general, this is a well-designed study and will provide value to the community. The reason behind the weak accept is due to limitations in the validation, mainly: 1) other tasks could be evaluated other than catheter segmentation; 2) only one comparative method was implemented; 3) evaluated only on a single dataset.

**Questions To Address In The Rebuttal:**

Two main questions:
1) How would you practically choose a suitable noise parameter $\rho$? Is it application dependent? I think this should be discussed in a sentence.
2) What is the impact of the NeRV neural network architecture on the segmentation and re-identifcation performance? What is also the impact of the activation functions?

---

> ### Author Response · Authors · 2024-03-18
>
> We thank the reviewer for their positive remarks and have addressed the remaining concerns (see also general comment). As suggested we had applied our method with the same hyperparameters to a different dataset - digitally reconstructed radiographs from thorax CT for re-identificaiton - and a different downstream task - clavicle segmentation as illustrated in our GitHub repository, which yielded very similar results and will be mentioned in the revised appendix. Furthermore we extended the choice of baseline methods by deformation-based obfuscation and mixup-privacy with detailed explanation in the general comment.
>
> Regarding the direct questions for the rebuttal: 1) Yes the selection of the noise parameter would be application specific. We will provide with the following suggestion for finding a good balance in practice: "Choosing $\rho$ depends on the intended use case and privacy-risk assessment. Since pre-trained task- and re-identification models can be quickly evaluated with different $\rho$s for a given NeRV model this provides a first indication of suitable choices. A re-training of both networks is, however, required to validate this assessment." 2) We mentioned in the second paragraph of Section 4 that a NeRV with only 500k trainable parameters per batch led to underperforming image reconstruction, hence a less-compact architecture was chosen - which ensured a PSNR of approx. 40 dB. Contrary to conventional coordinate MLPs that take xy-coordinates as input and require either sinusoidal activations (SIREN) or Fourier feature encodings, our approach already uses a learnable patient embedding and we did not see gains of modulating the activation function.

---

### Author Response · Authors · 2024-03-18
**Summary comment to all reviewers**

We would like to thank all reviewers for their very insightful comments and overall appreciation of our paper including the importance and practicality of the sharing of datasets (Wxho), interesting idea with well designed description (QorR and Wxho), its novelty (iTVz) and open source code (iTVz) along with our mentioning of limitations (QorR). The assessment of (iTVz) also points to an **important potential for clarification** regarding the re-training of re-identification attacks with knowledge of the NeRV-based obfuscation strategy. **This is certainly important and we had already done it but not mentioned clearly enough and hence a major concern of (iTVz) should be alleviated**.

Furthermore, all reviewers suggest to perform additional experiments for improved validation across datasets and tasks and in comparison to state-of-the-art (SotA). There are certain restrictions (see limitation section in paper) that we wish to maintain for our privacy-preserving data sharing experiments: the work of Packhäuser et al. e.g. assumes access to an already trained task model to perform obfuscation and some aspects of Kim et al. are set within a scenario where not all data from multiple providers is publicly shared. Nevertheless, **we designed two additional new experiments** inspired by the SotA of Kim et al. and Packhäuser et al.: **1) deformation-based obfuscation and 2) mix-up privacy;** details of which can be found in the revised appendix. For 1) we create smooth, invertible local deformations to obscure the identity and do indeed see a 20% drop in top5 re-identification risk. However, when re-training the attack model with knowledge about deformations (online augmentations) the risk increases again by 10% making it 30% less safe than NeRV with \rho=0.06. For 2) the scenario of sharing mix-up versions of images and labels without requiring a dual client-server training setup is more challenging (but also possible) and leads to a great reduction of privacy risks (by a factor of 2 or 4 in our tests): we could, however, not avoid a substantial drop in segmentation accuracy to about 40% Dice score for 4-fold mix-up (lower than the strongest pixelation strategy). We attribute this to the fact that jointly training on multiple images with similarly looking thin foreign objects (catheters, tubes, electrodes) is less stable than brain segmentation. Future work could strengthen a combination of these orthogonal strategies.

In addition, we would like to **point to two further experiments that were already included in our public GitHub repository** in time for review. They demonstrate the transfer of our method with same hyper-parameters onto a slightly different modality and a new downstream segmentation task. 1) We created DRRs (digitally reconstructed radiographs) of a public paired thorax CT dataset and evaluated the gains in re-identification risk (top5) reduction from 72.92% to 53.12% using NeRV (with \rho=0.08) 2) Due to the absence of fine-grained structures in large-scale X-ray databases (e.g. https://github.com/ngaggion/CheXmask-Database only provides masks for lungs and heart) we evaluated the possibility of segmenting the clavicles in the Montgomery County CXR dataset. This also led to satisfactory quantitative results of 81% Dice for qualitatively strongly obfuscated images. The repository also contains code for data preparation to further extend the experiments once more comprehensive data becomes available.

---

> ### Author Response · Authors · 2024-03-27
>
> We would like to highlight that all reviewers valued our contribution with respect to importance of the addressed privacy research challenge and acknowledged is as technically novel and interesting. There was constructive criticism about our initial focus on a single dataset, comparison method and downstream task, which we have alleviated to some degree by adding additional experiments and ablations.
>
> Reviewer iTVz, however, seems to be fundamentally opposed to our different research hypothesis that could be seen orthogonal to approaches of e.g. Kim et al. Their main reasoning is that they believe privacy-preserving image sharing should be performed using **visually** strongly obfuscated images (making references to our Fig. 4 which shows the same scan with different levels of obfuscation and prior work of Kim et al.), but carry the risk of being reverted by a strong attack. Whereas our approach introduces subtle variations by mixing small details across patients using a new neural field-based k-anonymity approach, which are harder to reverse for the siamese network as they stem from real other scans.
>
> Despite the strong opposition of one reviewer against such an alternate approach, we nevertheless feel that the MIDL community would benefit from a presentation of our work as it lies at the core of conferences to also enable a discussion of methods that are seen as controversial by some.

---

### Meta-Review · Area_Chair_x9s4 · 2024-04-04

**Recommendation:** Accept (Poster)
**Confidence:** 4

**Metareview:**

The paper describes a new approach to privacy-preserving data sharing. There has been an active discussion between the authors and the reviewers, and reviewer ratings vary. In particular, some of reviewer iTVz's concerns remain. I agree with the reviewer that the work appears in places preliminary, and additional experiments could strengthen the paper (such as tumor detection suggested by reviewer QorR ). However, I also believe the work would be of interest to the MIDL community and might raise interesting discussions on this work and privacy-preserving data-sharing approaches in general. As such, I believe it would have added value to the conference.

---

### Decision · Program_Chairs · 2024-04-06

Accept (Poster)